# MOBT Alleviates Pulmonary Fibrosis through an lncITPF–hnRNP-l-Complex-Mediated Signaling Pathway

**DOI:** 10.3390/molecules27165336

**Published:** 2022-08-22

**Authors:** Pan Xu, Haitong Zhang, Huangting Li, Bo Liu, Rongrong Li, Jinjin Zhang, Xiaodong Song, Changjun Lv, Hongbo Li, Mingwei Chen

**Affiliations:** 1Department of Respiratory and Critical Care Medicine, The First Affiliated Hospital of Xi’an Jiaotong University, Xi’an Jiaotong University, Xi’an 710061, China; 2Department of Respiratory and Critical Care Medicine, Binzhou Medical University Hospital, Binzhou Medical University, Binzhou 256603, China; 3Department of Cellular and Genetic Medicine, School of Pharmaceutical Sciences, Binzhou Medical University, Yantai 264003, China

**Keywords:** pulmonary fibrosis, lncITPF, hnRNP L, MEF2c

## Abstract

Pulmonary fibrosis is characterized by the destruction of alveolar architecture and the irreversible scarring of lung parenchyma, with few therapeutic options and effective therapeutic drugs. Here, we demonstrate the anti-pulmonary fibrosis of 3-(4-methoxyphenyl)-4-oxo-4*H*-1-benzopyran-7-yl(αS)-α,3,4-trihydroxybenzenepropanoate (MOBT) in mice and a cell model induced by bleomycin and transforming growth factor-β1. The anti-pulmonary fibrosis of MOBT was evaluated using a MicroCT imaging system for small animals, lung function analysis and H&E and Masson staining. The results of RNA fluorescence in situ hybridization, chromatin immunoprecipitation (ChIP)-PCR, RNA immunoprecipitation, ChIP-seq, RNA-seq, and half-life experiments demonstrated the anti-pulmonary fibrotic mechanism. Mechanistic dissection showed that MOBT inhibited lncITPF transcription by preventing p-Smad2/3 translocation from the cytoplasm to the nucleus, resulting in a reduction in the amount of the lncITPF–hnRNP L complex. The decreased lncITPF–hnRNP L complex reduced MEF2c expression by blocking its alternative splicing, which in turn inhibited the expression of MEF2c target genes, such as TAGLN2 and FMN1. Briefly, MOBT alleviated pulmonary fibrosis through the lncITPF–hnRNP-l-complex-targeted MEF2c signaling pathway. We hope that this study will provide not only a new drug candidate but also a novel therapeutic drug target, which will bring new treatment strategies for pulmonary fibrosis.

## 1. Introduction

Pulmonary fibrosis is a chronic progressive fibrotic interstitial pneumonia characterized by irreversible scarring of the lung parenchyma and restrictive ventilatory dysfunction [1] in which various lung cells, including alveolar epithelial, fibroblast, macrophage, and endothelial cells participate [2]. Accordingly, pulmonary fibrosis is involved in a variety of molecular mechanisms. Abnormal alveolar epithelial wound healing is regarded as the initial factor. Then, the aberrant cells release growth factors and profibrotic cytokines that create a microenvironment for the transformation of fibroblasts into myofibroblasts. Excessive activation and proliferation of myofibroblasts results in a plethora of deposition in the extracellular matrix, which decreases lung compliance and causes gas exchange disorders [3,4]. In 2014, the FDA approved two antifibrotic drugs for the treatment of pulmonary fibrosis: nintedanib and pirfenidone. Although several studies have shown that both drugs can slow down the decline rate of forced vital capacity (FVC) in patients with idiopathic pulmonary fibrosis (IPF), they cannot reverse the fibrosis process and prolong the survival time of patients [5,6]. Data on the side effects of these drugs, such as liver function damage, gastrointestinal discomfort, and photosensitivity, continue to emerge in recent research [7]. Although several studies have demonstrated the potential benefits of certain drugs, official statements for these drugs have strongly advised extreme caution [1,8]. Therefore, this condition highlights the difficulties of the pharmacological treatment of pulmonary fibrosis.

Fibroblast-to-myofibroblast differentiation is one of the most important pathogeneses associated with pulmonary fibrosis. The primary pathological manifestations are uncontrolled proliferation and migration of myofibroblasts. So, inhibition of fibroblast-to-myofibroblast differentiation is an attractive approach to block the fibrosis process [9]. Several compounds, including interfering peptide and interference RNA, have been described as fibrotic inhibitors capable of inhibiting fibroblast-to-myofibroblast differentiation. According to Zhou et al., aucubin restrains myofibroblast proliferation and differentiation, thus protecting against bleomycin (BLM)-induced mouse pulmonary fibrosis [10]. Dong et al. developed a follistatin-like 1 neutralizing antibody named 22B6 mAb and demonstrated that it attenuates pulmonary fibrosis through blocking fibroblast differentiation and extracellular matrix production [11]. Xu et al. reported that siRNA, circANKRD42, alleviates pulmonary fibrosis by inhibiting fibroblast differentiation into myofibroblasts and myofibroblast proliferation and migration [12]. However, advanced pulmonary fibrosis treatment research, such as drug synthesis and mechanism, is still insufficient.

Long noncoding RNA (lncRNA), as a diagnostic marker or therapeutic target in numerous diseases, is a critical regulator of gene function and regulation [13]. According to Yang et al., the interaction of lncRNA OIP5-AS1 with myocyte enhancer factor-2c (MEF2c) promotes myoblast differentiation into myotubes during muscle development and regeneration after injury [14]. The lncXIST–TRIM28 complex maintains the biological function of X-inactivation in B cells [15]. In a previous study, we first revealed the differentially expressed profiles of lncRNAs in BLM-induced pulmonary fibrosis using microarray analysis and demonstrated that the regulatory mechanism of lncITPF on pulmonary fibrosis is to form an lncITPF–hnRNP L complex to facilitate pulmonary fibrosis [16,17]. However, whether the lncITPF–hnRNP L complex can be a drug target is yet to be explored.

3-(4-Methoxyphenyl)-4-oxo-4*H*-1-benzopyran-7-yl(αS)-α,3,4-trihydroxybenzenepropanoate (MOBT) is a novel compound synthesized by our group by using formononetin and danshensu as raw materials (Figure 1A). In the present study, we demonstrated that the anti-pulmonary fibrosis of MOBT in mice and a cell model induced by BLM and transforming growth factor-β1 (TGF-β1), respectively. Further mechanistic study elucidated that the regulatory mechanism of MOBT is through the lncITPF–hnRNP-l-complex-targeted MEF2c signaling pathway.

## 2. Results

### 2.1. MOBT Prevented BLM-Induced Lung Fibrosis in Mice

Animal experiments were performed to evaluate the anti-fibrotic effect MOBT on BLM-induced pulmonary fibrosis in mice (Figure 1A). The MicroCT imaging system for small animals demonstrated that both lungs of mice in the BLM group had diffuse reticular blurring, whereas MOBT alone had no obvious changes compared with the sham group. The BLM + MOBT treatment group significantly reduced the lesion range of reticular changes compared with the BLM group. (Figure 1B). Weight monitoring showed that MOBT effectively slowed down the weight loss of BLM-treated mice (Figure 1C). Lung function analysis demonstrated that the MOBT treatment improved the FVC compared with the BLM-treated mice (Figure 1D). H&E and Masson staining revealed that the BLM group had more collagen deposition and a thicker alveolar wall than the sham group, whereas the MOBT alone group was similar to the sham. The BLM + MOBT treatment attenuated collagen deposition and had a significant therapeutic effect (Figure 1E). The Western blot result revealed that MOBT distinctly decreased the expression of fibrotic markers, including α-SMA, vimentin, and collagen I, compared with the BLM group (Figure 1F). Meanwhile, MOBT also inhibited the expression of cytokines related to fibrosis, such as TGF-β1, CTGF, and VEGF (Figure 1G). Immunofluorescence and immunohistochemistry results confirmed that the lung mesenchyme was positively stained for α-SMA in the BLM group, indicating the appearance of fibroblastic foci. The MOBT treatment reduced α-SMA expression and improved the fibrotic status (Figure 2). The above-mentioned results indicate the anti-pulmonary fibrotic ability of MOBT in vivo.

### 2.2. MOBT Prevented Pulmonary Fibrosis via Inhibiting lncITPF Transcription

lncRNA is becoming a new therapeutic target [18,19]. lncITPF was selected for further study because it is a therapeutic target for pulmonary fibrosis [16]. To further explore the regulatory mechanism of MOBT on lncITPF, TGF-β1 was used to activate MRC-5 cells to create a cell model of pulmonary fibrosis. Western blot results demonstrated that TGF-β1 increased the expression of fibrotic markers, such as collagen I, vimentin, and α-SMA, whereas MOBT decreased their expression (Figure 3A), which indicated the anti-pulmonary fibrotic ability of MOBT in vitro. Next, the regulatory mechanism of MOBT on lncITPF was further explored in the TGF-β1-activated cell model. Quantitative real-time polymerase chain reaction (qRT-PCR) data clarified that MOBT inhibited the lncITPF expression level (Figure 3B). Gain- and loss-of-function assays were performed by transfection with lncITPF-overexpressing vector (recombinant plasmid, RP) and specific lncITPF RNA smart silencer (si-lncITPF). The rescue experiment for real-time cellular analysis (RTCA) demonstrated that MOBT inhibited the proliferation and migration of cells treated with TGF-β1, but lncITPF overexpression promoted the proliferation and migration and reversed the effect of MOBT (Figure 3C,D). The result indicates that the therapeutic effect of MOBT depends on lncITPF.

The regulatory mechanism of MOBT on lncITPF was clarified through RNA fluorescence in situ hybridization (RNA-FISH), half-life of lncITPF, phosphorylation assay, and chromatin immunoprecipitation (ChIP)-PCR experiments. The cellular location of the gene determines the regulatory mechanism, such as transcriptional regulation in the nucleus and translational regulation in the cytoplasm. Accordingly, the location of lncITPF was tested under TGF-β1/MOBT treatment by single-molecule RNA-FISH. The images displayed that lncITPF primarily existed in the nucleus. The lncITPF location did not change with or without MOBT/TGF-β1 treatment (Figure 3E). Accordingly, we inferred that MOBT regulated lncITPF transcription. Then, we questioned how MOBT regulated lncITPF transcription. TGF-β1/Smad2/3 was proven to be the upstream signaling pathway of lncITPF in pulmonary fibrosis. When Smad2/3 receives the signal transmitted from TGF-β1, Smad2/3 translocates from the cytoplasm to the nucleus via phosphorylation to control its target gene transcription [20]. Consequently, the effect of MOBT on transcription factor Smad2/3 was detected. The phosphorylation of Smad2/3 assay proved that MOBT repressed Smad2/3 phosphorylation, indicating that MOBT blocked the translocation of Smad2/3 from the cytoplasm to the nucleus (Figure 3F). ChIP-PCR detection was performed to analyze the enrichment of Smad on the promoter of lncITPF under the MOBT action. The result verified that MOBT reduced the Smad2/3 enrichment in the lncITPF promoter (Figure 3G), indicating that MOBT blocked the transcriptional function of Smad2/3 for lncITPF. The half-life experiment confirmed that MOBT reduced the half-life of lncITPF compared with the TGF-β1-treated group (Figure 3H). The above findings illustrate that MOBT reduces the amount of Smad2/3 phosphorylation, resulting in the blockage of p-Smad2/3 translocation from the cytoplasm to the nucleus and inhibiting lncITPF transcription.

### 2.3. MOBT Downregulated lncITPF–hnRNP-l-Complex-Targeted MEF2c Splicing

The regulatory mechanism of lncITPF in pulmonary fibrosis is to form an lncITPF–hnRNP L complex to facilitate pulmonary fibrosis. Accordingly, we explored the target gene of the lncITPF–hnRNP L complex. hnRNP L is an alternative splicing factor and effects its target gene splicing. So, an RNA-binding protein immunoprecipitation (RIP) experiment was performed to discover the target gene of hnRNP L. The result uncovered that hnRNP L bound to pre-mMEF2c and MOBT reduced the binding of hnRNP L with MEF2c (Figure 4A). Then, the MEF2c expression was analyzed by Western blot in the MRC-5 cells treated with TGF-β1 for different times to confirm the change in MEF2c in the pulmonary fibrosis cell model. The result elucidated that MEF2c gradually increased with the extension of TGF-β1 action time. The MEF2c expression was highest after 72 h of TGF-β1 treatment (Figure 4B), so the condition was selected for further studies. The cells were treated with 5 ng/mL TGF-β1 for 24 h, then co-treated with 10 μg/mL MOBT for 48 h. The Western blot result illustrated that MOBT attenuated the MEF2c expression at the protein level (Figure 4C). Immunofluorescence images depicted that MOBT treatment decreased the MEF2c expression compared with the TGF-β1 treatment (Figure 4D). The MEF2c expression was increased in the IPF patient’s pulmonary tissues compared with the normal subjects (Figure 4E), indicating that the study of MOBT targeted MEF2c has clinical value in medical treatment. The stability of the MEF2c protein was measured under MOBT action using a cycloheximide experiment to further elucidate the mechanism of MOBT on MEF2c. The half-life of MEF2c was 4.36, 11.65, and 5.96 h in the normal, TGF-β1, and TGF-β1 + MOBT groups, respectively, indicating that MEF2c stability was enhanced by TGF-β1 but reduced by MOBT treatment (Figure 4F). Next, the MEF2c protein level was evaluated after the cells were co-transfected with TGF-β1 and with overexpression or knockdown lncITPF. The result shows that lncITPF overexpression increased the MEF2c level, but knockdown of lncITPF caused its reduction (Figure 5A,B). The rescue experiment of Western blot demonstrated that MEF2c decreased under MOBT treatment in TGF-β1-activated cells. Overexpression of lncITPF promoted the MEF2c expression and reversed the role of MOBT, indicating that the inhibition of MOBT on MEF2c depended on lncITPF (Figure 5C). Accordingly, we explored the effect of lncITPF–hnRNP L on MEF2c under MOBT action. The rescue experiment demonstrated that si-hnRNP L repressed MEF2c expression and reversed the joint action of MOBT and overexpression lncITPF (Figure 5D), indicating that the effect of MOBT on MEF2c depended on the lncITPF–hnRNP L complex. The above findings illustrate that MOBT downregulates MEF2c expression through the lncITPF–hnRNP L complex.

### 2.4. MOBT Downregulated the Target Genes of MEF2c

MEF2c is a key transcription factor [21]. We explored which target genes were regulated by MEF2c to control lung fibrogenesis. A ChIP-seq experiment was performed to identify genes binding with MEF2c (Figure 6A). RNA-seq was performed to analyze the differentially expressed genes (Figure 6B). The ChIP-seq data shows a total of 2704 differentially expressed genes in the TGF-β1 group, with 1136 upregulated and 1568 downregulated genes compared with the normal control. The finding indicates that MEF2c controlled these target genes by binding to the cis-acting elements in their promoter regions. The RNA-seq data displayed a total of 1483 differentially expressed genes in the TGF-β1 group, with 407 upregulated and 1076 downregulated genes compared with the normal control. The combined analysis demonstrated a total of 76 target genes in both ChIP-seq and RNA-seq data, with 11 upregulated and 65 downregulated genes (Figure 6C). The KEGG analysis revealed that the 76 target genes regulated by MEF2c were significantly enriched in the Hippo, Wnt, and MAPK signaling pathways, which are involved in pulmonary fibrogenesis via fibroblast differentiation, migration, proliferation, and extracellular matrix (Figure 6D). The 44 significant differentially expressed genes were selected from 76 differentially expressed genes and listed in the hierarchical clustering of RNA-seq (Figure 6E). Finally, the binding sites for MEF2c in the genes of transgelin 2 (TAGLN2) and formin 1 (FMN1) were further analyzed (Figure 6F). qRT-PCR confirmed the sequencing data that TAGLN2 and FMN1 decreased under MOBT treatment compared with those in TGF-β1 group (Figure 7A). The RIP experiment proved that MEF2c bound to TAGLN2 and FMN1 (Figure 7B). The Western blot result verified that MOBT inhibited TAGLN2 and FMN1 at the protein level (Figure 7C). The above findings suggest that MOBT prevents pulmonary fibrosis by regulating the target genes of MEF2c, such as TAGLN2 and FMN1.

## 3. Discussion

This study proved that MOBT alleviated pulmonary fibrosis through downregulating the lncITPF–hnRNP-l-complex-targeted MEF2c signaling pathway in vitro and in vivo. The results of the RNA-FISH, ChIP-PCR, RIP, ChIP-seq, RNA-seq, and half-life experiments elucidated the anti-pulmonary fibrotic mechanism. Mechanistic dissection revealed that MOBT inhibited lncITPF transcription by preventing p-Smad2/3 translocation from the cytoplasm to the nucleus, resulting in a reduced amount of lncITPF–hnRNP L complex. The decreased lncITPF–hnRNP L complex reduced MEF2c expression by blocking its alternative splicing, which in turn inhibited the expression of MEF2c target genes, such as TAGLN2 and FMN1 (Figure 8).

In recent years, lncRNAs have received increasing attention in the fields of life science, medicine, and pharmacology because of their conservation and high tissue-specificity [22,23,24]. Cis-acting lncRNAs regulate expression of genes that are located on the same chromosome, whereas trans-acting lncRNAs regulate genes on other chromosomes [25]. They play important functions in the spatial conformation of chromosomes, RNA transcription, pre-mRNA splicing, mRNA degradation, and translation by interacting with proteins [13,26]. For example, the lncIAPF–HuR complex inhibits autophagosome fusion with lysosomes to block autophagy, resulting in pulmonary fibrogenesis [27]. lncHOTAIR regulates tumor cell proliferation by binding to Y-Box Protein-1 and promoting its nuclear translocation, which stimulates Y-Box Protein-1 downstream targets PCK2 and PDGFRβ [28]. Accordingly, lncRNAs have become therapeutic targets in many diseases, such as pulmonary, renal, liver, and cardiac fibrosis [9,29]. lncMER52A can serve as a diagnostic and prognostic marker for hepatocellular carcinoma [30]. lncRAPIA may become a new prevention and treatment target for advanced atherosclerosis [31]. However, these studies mostly focused on gene therapy and paid little attention to drug treatment. Numerous herbal compounds, such as curcumin, camptothecin, resveratrol, quercetin, genistein, 3,3′-diindolylmethane, and epigallocatechin-3-galate, exhibit anticancer effects through controlling lncRNA expression [32]. lncRNA-mRNA ChIP analysis uncovered the action of *Caulophyllum robustum* Maxim, which can play a part in rheumatoid arthritis via lncRNA-mediated signing pathways, including tumor necrosis factor, chemokine, and Toll-like receptor signaling pathways [33]. These studies simply screened the differentially expressed lncRNAs under the action of drugs. However, the drug mechanism mediated by lncRNA remains obscure. This work found that MOBT blocked MEF2c alternative splicing to prevent pulmonary fibrosis by reducing the formation of the lncITPF–hnRNP L complex, and that the lncITPF–hnRNP L could be a therapeutic drug target in pulmonary fibrosis.

hnRNP L is an alternative splicing factor. RNA alternative splicing is a basic post-transcriptional processing that allows a gene to produce many different transcripts and proteins [34]. Tan et al. indicated that the lncSNHG1–hnRNP L complex can boost prostate cancer growth and metastasis through impairing the translation of protein E-cadherin and activating epithelial–mesenchymal transition progress [35]. Fan et al. discovered that lncRBAT1 accelerates tumorigenesis through directly recruiting hnRNP L to the E2F3 promoter and cis-activating E2F3 expression [36]. Our results reveal that the lncITPF–hnRNP L complex bound to MEF2c pre-mRNA to enhance its alternative splicing. The MOBT treatment inhibited MEF2c pre-mRNA alternative splicing to reduce MEF2c expression via decreasing the lncITPF–hnRNP L complex.

MEF2c belongs to the MEF2 transcription factor family, which is expressed in a variety of cells and associated with neurodevelopmental disease, heart failure, and vascular inflammation [37,38,39]. MEF2c initiates the transcription of target genes and induces the cells to develop a series of adaptive responses. It can direct fibroblast reprogramming and correlates with fibrosis in vivo. According to Zhang et al., salvianolic acid B prevents hepatic stellate cell activation and hepatic fibrogenesis by downregulating the MEF2c signaling pathway [40]. Our RIP, qRT-PCR, and Western blot data confirmed that MEF2c bound to the promoters of TAGLN2 and FMN1 and promoted the transcription of these target genes. These target genes belong to the Hippo, Wnt, and MAPK signaling pathways, which can activate myofibroblast differentiation, proliferation, migration, and ECM deposition to promote pulmonary fibrosis. Meanwhile, MOBT treatment inhibited TAGLN2 and FMN1 through downregulating lncITPF–hnRNP-l-complex-targeted MEF2c.

In conclusion, our study shows that MOBT alleviated pulmonary fibrosis through the lncITPF–hnRNP-l-complex-targeted MEF2c signaling pathway. We hope that this study provides not only a new drug candidate but also a novel therapeutic drug target that will bring new treatment strategies for pulmonary fibrosis.

## 4. Materials and Methods

### 4.1. Patient Samples

Lung tissue samples of IPF patients were obtained by transbronchial lung biopsy. Control non-pulmonary fibrosis specimens were obtained from paracancerous tissues of primary lung cancer patients. All patients provided written informed consent, and ethical consent was granted by the Committee of Binzhou Medical University Hospital (Ethical Research approval No. 2019-G049-01).

### 4.2. Animal Model and Ethics Statement

C57BL/6 male mice were purchased from Jinan Pengyue Experimental Animal Breeding Company (Jinan, China). The mice were randomly divided into sham, BLM, and BLM + MOBT groups (10 mice in each group). An amount of 5 mg/kg BLM was sprayed into the lungs of mice by a Penn-Century MicroSprayer (Penn-Century Inc., Philadelphia, PA, USA) in the BLM and BLM + MOBT groups. From the eighth day, the BLM + MOBT group was orally administered 30 mg/kg MOBT every day. The sham group was sprayed the same dose of normal saline into the lung as BLM. On the 28th day, the lung changes of all mice were evaluated by a MicroCT imaging system for small animals (PerkinElmer, Waltham, MA, USA). Then, lung samples were collected, processed, and stored for the following experiments. Experiments on the animals were approved by the Animal Experiments Ethics Committee of Binzhou Medical University (Ethical Research approval No. 2017-02).

### 4.3. Cell Model and Treatment

Human embryonic lung fibroblast MRC-5 cell line was purchased from the American Type Culture Collection (ATCC, CCL-171™, Rockefeller, MD, USA). The cells were cultured in advanced minimum essential medium (Gibco, 11090081, Grand Island, NY, USA) containing 10% FBS (Gibco, 10270106, Grand Island, NY, USA) in a 37 °C, 5% CO_2_ incubator. Cell samples were divided into three groups: normal, TGF-β1, and TGF-β1 + MOBT groups. The TGF-β1 + MOBT group was first treated with 5 ng/mL TGF-β1 (Gibco, PHG9202, Grand Island, NY, USA) for 24 h and then co-treated with 10 μg/mL MOBT for 48 h.

### 4.4. Western Blot

Western blot was conducted with the standard protocols established by our laboratory, specific steps were referred to published articles [12]. Antibodies are listed in Table 1.

### 4.5. H&E and Masson Staining

The harvested lung tissues were immediately fixed in 4% paraformaldehyde (Meilunbio, MA0192, Dalian, China) overnight, then dehydrated and embedded in paraffin. Sections (4 μm) were stained with H&E (Novland, IH-017 and IH-018, Shanghai, China) or a modified Masson’s trichrome staining kit (Solarbio, G1345, Beijing, China). Images were obtained under the microscope.

### 4.6. Immunofluorescence Observation

Lung tissue paraffin sections were dewaxed according to the manufacturer’s instructions, incubated with 0.3% TritonX-100 (Sinopharm, 30188928, Shanghai, China) for 8 min, blocked with 10% goat serum (Solarbio, SL038, Beijing, China) for 50 min, then incubated with the antibody of α-SMA (1:200, Abcam, Ab7817, Cambridge, UK) at 4 °C overnight. On the second day, sections were rinsed with 1×PBS solution (Sparkjade, CR0013, Jinan, China) and incubated with anti-IgG Fluor (1:200, Affinity, S0006, Beijing, China) for 1 h. The nuclei were stained with DAPI (1:400, Sigma, D9542, St. Louis, MO, USA). After 1×PBS washing, tissue sections were sealed with neutral glycerin and photos taken in an automatic living cell fluorescence microscopic imaging system (Invitrogen, EVOS M5000, Carlsbad, CA, USA).

### 4.7. Half-Life of lncITPF Analysis

Cells (1 × 10^6^/mL) were seeded in a 6 cm petri dish. A total of 2 mL serum-free medium containing 5 μg/mL actinomycin D (Aladdin, A113142, Shanghai, China) was added into the cell samples. Total RNA was extracted and lncITPF levels were measured by qRT-PCR at different time points. GAPDH was used as an internal standard.

### 4.8. ChIP-PCR

A ChIP-PCR assay was performed by using the SimpleChIP^®^ enzymatic chromatin IP kit (Cell Signaling Technology, #9002, Danvers, MA, USA) according to the manufacturer’s instructions. Briefly, cell samples were crosslinked with 1% formaldehyde (Macklin, F809702, Shanghai, China) for 10 min at room temperature. Then, 2.5 M glycine (Meilunbio, MB4166, Dalian, China) was added to for 5 min terminate crosslinking. Chromatin was immunoprecipitated with anti-Smad2/3 (Cell signaling technology, 8685, Danvers, MA, USA) or IgG (Cell signaling technology, 2729, Danvers, MA, USA) overnight. The antibody/antigen complexes were recovered with protein G agarose beads (Cell signaling technology, 9007, Danvers, MA, USA) for 2 h at 4 °C. After two sequential elutions, 0.2 M NaCl was added to the eluent at 65 °C overnight to decrosslink. The immunoprecipitated DNA was collected and then tested by PCR. The primers are listed in Table 2.

### 4.9. RNA-FISH

lncITPF, U6, and 18S FISH probes were synthesized by Guangzhou RiboBio Co., Ltd. The experiment was performed with the Ribo lncRNA FISH Probe Mix according to the manufacturer’s protocol (RiboBio, C10910, Guangzhou, China). Cells were inoculated on a circular slide. When the cell density reached 50–60%, 4% paraformaldehyde (Meilunbio, MA0192, Dalian, China) was added. Then, cell samples were washed with 1 × PBS (Sparkjade, CR0013, Jinan, China) and punched with 0.3% TritonX-100 (Sinopharm, 30188928, Shanghai, China) at 4 °C for 3 min. Permeate solution was washed away with 1 × PBS and 200 μL pre-hybridization solution was added for 30 min. Subsequently, lncITPF, U6, and 18S FISH Probe Mix were added to the cell samples in a hybridization oven at 37 °C overnight. On the second day, the hybridization probe solution was aspirated at 42 °C and washed with SSC (Solarbio, S1030, Beijing, China). Then DAPI (Sigma, D9542, St. Louis, MO, USA) solution was added for 6 min. Finally, fluorescence was observed using a laser confocal microscope.

### 4.10. RTCA

Cells (5 × 10^4^/mL) were seeded in an E-plate (Agilent, 5469830001, Santa Clara, CA, USA) for proliferation analysis and a CIM plate (Agilent, 05665817001, Santa Clara, CA, USA) for migration analysis. The CIM plate contains an upper chamber and a lower chamber. A total of 30 μL serum-free medium and 165 μL serum medium was added into the upper or lower chambers, respectively. The RTCA instrument automatically recorded the proliferation or migration curves. The cell index was calculated using RTCA software (ACEA Biosciences, San Diego, CA, USA).

### 4.11. RIP Analysis

An RIP assay was performed using an EZ-Magna RIP™ RNA-binding protein immunoprecipitation kit (Millipore, 17-701, Bill Ricard, MA, USA) according to the manufacturer’s instructions. Firstly, RIP lysis buffer was added to cell samples. Then hnRNP-L antibody (Abcam, ab264340, Cambridge, UK) or IgG antibody (Cell Signaling Technology, 2729, Danvers, MA, USA) conjugated with magnetic beads was added to the whole cell extract at 4 °C overnight. Next, beads were collected after incubation with protein A/G at 4 °C for 2 h and washed 5 times. Purified RNA was analyzed by qRT-PCR with MEF2c-pre-mRNA-specific primers. The primers are listed in Table 2.

### 4.12. RNA-Seq

A total of 2 μg RNA per sample was used as the initial material for the RNA sample preparations. Ribosomal RNA was removed using Ribo-off rRNA Depletion Kit (Nanjing, China). Subsequently, the sequencing libraries were generated following manufacturer recommendations with varied index label by NEBNext Ultra Directional RNA Library Prep Kit for Illumina (NEB, Ispawich, MA, USA). The RNA concentration of the library was measured using a Qubit RNA Assay Kit in Qubit 2.0 to preliminarily quantify and then dilute to 1 ng/μL. Insert size was assessed using the Agilent Bioanalyzer 2100 system (Agilent Technologies, Santa Clara, CA, USA) and qualified insert size was accurately quantified using the Taqman fluorescence probe of AB Step One Plus Real-Time PCR system (library valid concentration > 10 nM). The cluster generation and sequencing were performed on a Novaseq 6000 S4 platform, using a NovaSeq 6000 S4 Reagent kit V1.5.

### 4.13. ChIP-Seq

The ChIP assay was performed by Shandong Xiuyue Biotechnology Co., Ltd., (Jinan, China) according to the standard crosslinking ChIP protocol with modifications. Briefly, cells were harvested and crosslinked with 1% formaldehyde for 10 min at room temperature. After sonication, immunoprecipitation was performed with anti-MEF2c (Santa, sc-518152, 10 μL, Dallas, TX, USA). The immunoprecipitated complex was washed, and DNA was extracted and purified using a Universal DNA Purification Kit (TIANGEN, #DP214, Beijing, China). The ChIP-Seq library was prepared using original Ultra II DNA Library Kits (NEB, #E7645, Ispawich, MA, USA) according to the manufacturer’s instructions. For ChIP-seq, extracted DNA was ligated to specific adaptors followed by deep sequencing in the Illumina Novaseq 6000 using 150bp paired-end.

### 4.14. Statistical Evaluation

Statistical analyses were performed using SPSS version 19.0 software. Data are presented as the mean ± SD of at least three independent experiments. The unpaired Student’s *t*-test was used for experiments comparing two groups, whereas one-way ANOVA with the Student–Newman–Keuls post hoc test was applied for experiments comparing three or more groups. Statistical significance was considered at *p* < 0.05.

## Figures and Tables

**Figure 1 molecules-27-05336-f001:**
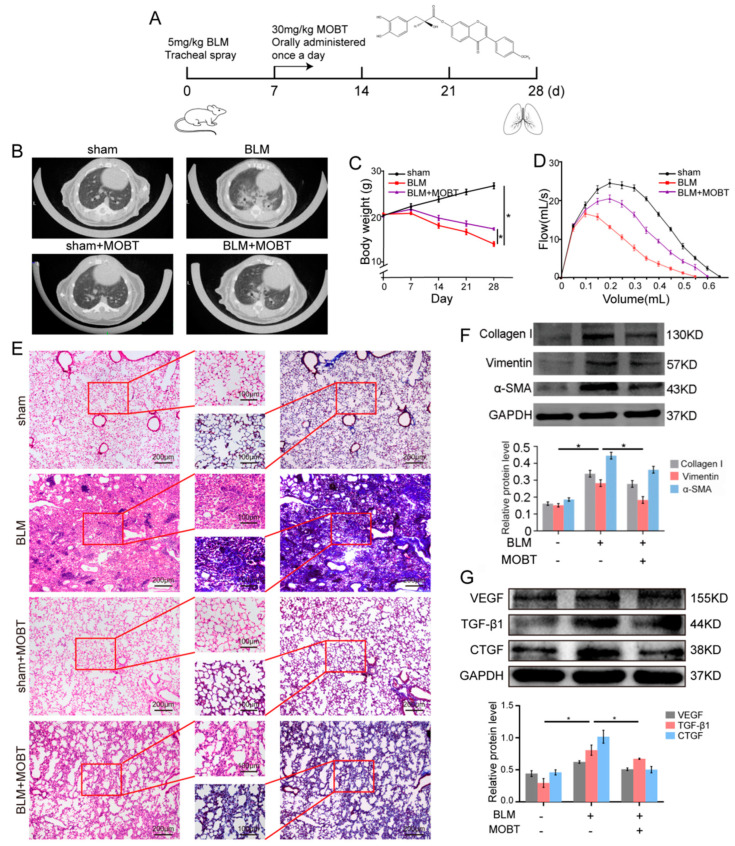
MOBT alleviated pulmonary fibrosis in vivo. (**A**) Molecular formula of MOBT and schematic illustration of MOBT administrated into mice. (**B**) The MicroCT imaging system for small animals displayed that the plain scan of the lung window in the sham group showed the texture of both lungs was clear, and no exudation or space occupying lesions were found in the lung parenchyma. The MOBT alone had no obvious changes compared with the sham group. However, both lower lungs of the mice in the BLM group showed diffused reticular blurring and interstitial changes. Meanwhile, the BLM + MOBT treatment group significantly reduced the lesion range of reticular changes. (**C**) The body weight monitoring revealed that mouse weight in the BLM group was significantly reduced compared with the sham group, whereas in the MOBT group it was increased compared with the BLM group. (**D**) The result of mice lung function showed that MOBT treatment improved the FVC. (**E**) H&E and Masson staining demonstrated that a portion of the alveolar structure was damaged, alveolar fusion, and collagen fibers were gathered in the lung of the BLM mice compared with the sham group, whereas MOBT alone group was similar to the sham. The BLM + MOBT treatment improved the alveolar structure and attenuated collagen deposition. (**F**) The Western blot demonstrated that MOBT inhibited the fibrotic protein expression, including collagen I, vimentin, and α-SMA compared with those in the BLM-treated group. (**G**) Western blot demonstrated that MOBT inhibited the expression of TGF-β1, CTGF, and VEGF, which are cytokines related to fibrosis, compared with those in the BLM group. Each bar represents mean ± SD, *n* = 6, * *p* < 0.05.

**Figure 2 molecules-27-05336-f002:**
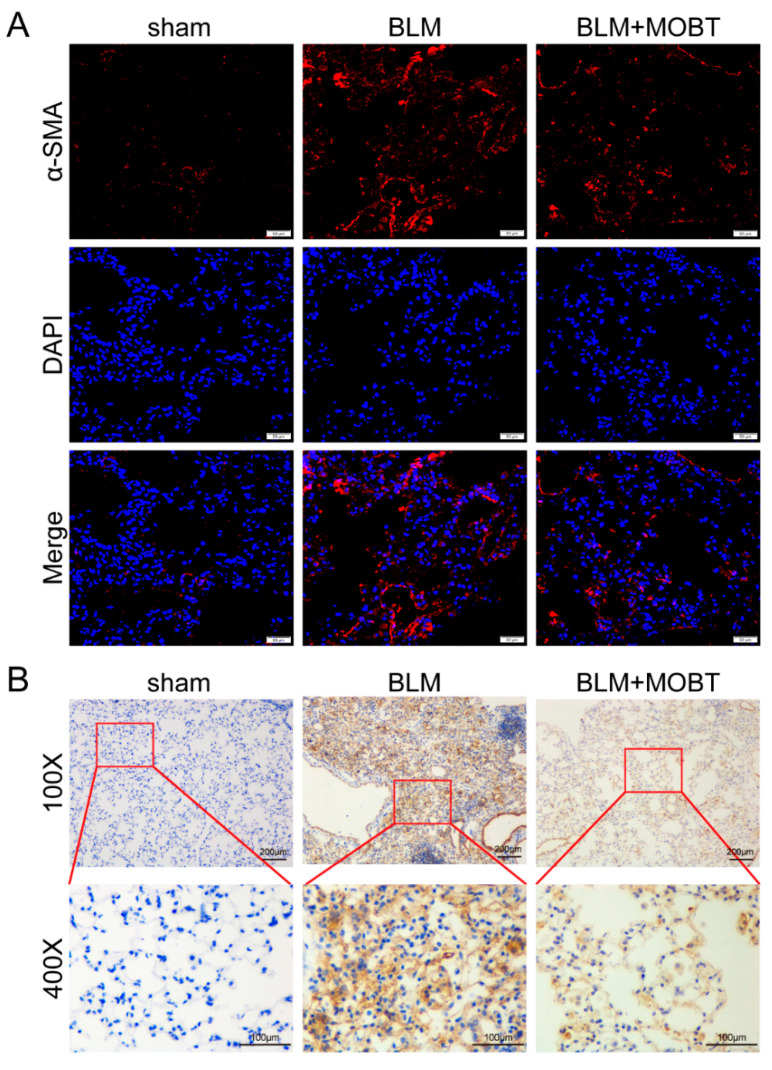
MOBT treatment reduced α-SMA expression. (**A**) The immunofluorescence result demonstrated that the BLM group expressed abundant α-SMA compared with the sham group, whereas the MOBT treatment significantly decreased the α-SMA expression. The red color indicates α-SMA. The blue color indicates nucleus. (**B**) The immunohistochemical result indicated that the sham group had distinct alveolar structures and thin alveolar walls. The alveolar structure in the BLM group was disordered and had thick alveolar walls. The MOBT treatment significantly improved the lung alveolar structure and decreased α-SMA deposition. The brown color indicates α-SMA.

**Figure 3 molecules-27-05336-f003:**
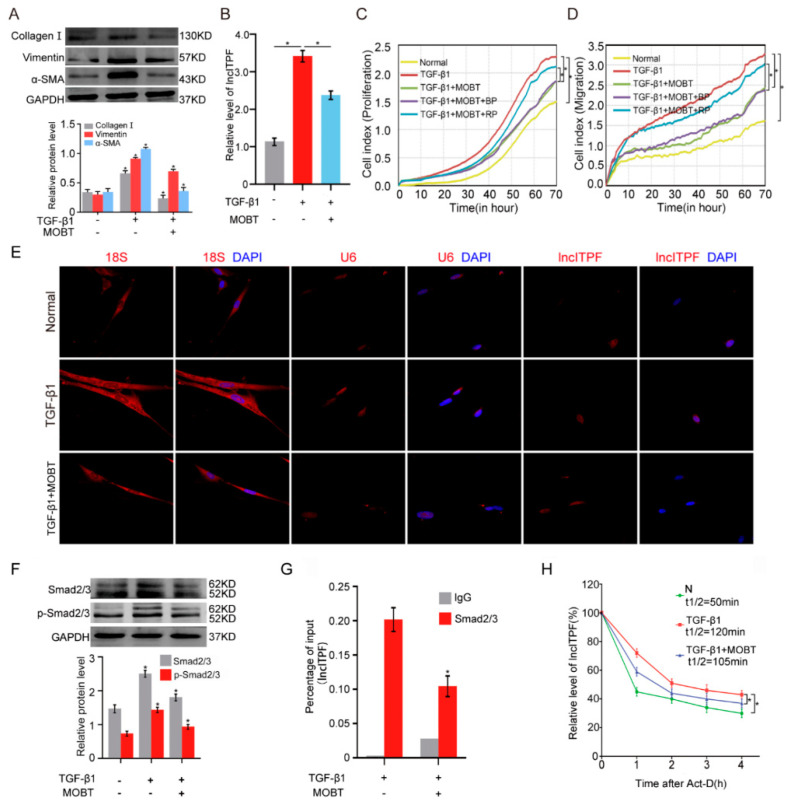
MOBT inhibited lncITPF transcription via reducing Smad2/3 phosphorylation. (**A**) The Western blot analysis displayed that TGF-β1 increased the expression of collagen I, vimentin, and α-SMA, whereas MOBT decreased their expression. (**B**) qRT-PCR was used to discover the lncITPF expression level in the normal, TGF-β1, and MOBT + TGF-β1 groups. The results demonstrated that TGF-β1 increased the lncITPF expression, whereas MOBT caused a reduction. (**C**) The xCELLigence real-time cell analysis (RTCA) system was used to analyze the proliferation of MRC-5 cells. Overexpression of lncITPF increased the proliferation of activated fibroblasts, similar to the TGF-β1 effect. MOBT reduced the proliferation of activated fibroblasts compared with the TGF-β1-treated group. (**D**) The RTCA system was used to assay the migration of MRC-5 cells. Overexpression of lncITPF increased the migration of activated fibroblasts, similar to the TGF-β1 effect. MOBT reduced the migration of activated fibroblasts compared with the TGF-β1-treated group. (**E**) Single-molecule RNA-FISH revealed that the lncITPF (red) was mostly located in the nucleus. MOBT treatment reduced the lncITPF expression and did not lead to its translocation from the nucleus to the cytoplasm. 18S RNAs were used as cytoplasmic localization markers. U6s were used as nuclear localization markers. DAPI was used for the staining of fixed cell DNA (blue). (**F**) The Western blot analysis demonstrated that MOBT inhibited Smad2/3 and p-Smad2/3 expression. (**G**) ChIP-PCR was used to evaluate the binding of Smad2/3 in the lncITPF promoter region under MOBT treatment. The results reflected that MOBT decreased Smad2/3 protein enrichment in the lncITPF promoter. (**H**) Actinomycin D rapidly decreased the expression of lncITPF, and the half-life of lncITPF was approximately 50 min. The combination of actinomycin D and TGF-β1 increased the half-life of lncITPF (120 min). Meanwhile, the combination of actinomycin D, TGF-β1, and MOBT decreased the half-life of lncITPF (105 min). Each bar represents mean ± SD, * *p* < 0.05.

**Figure 4 molecules-27-05336-f004:**
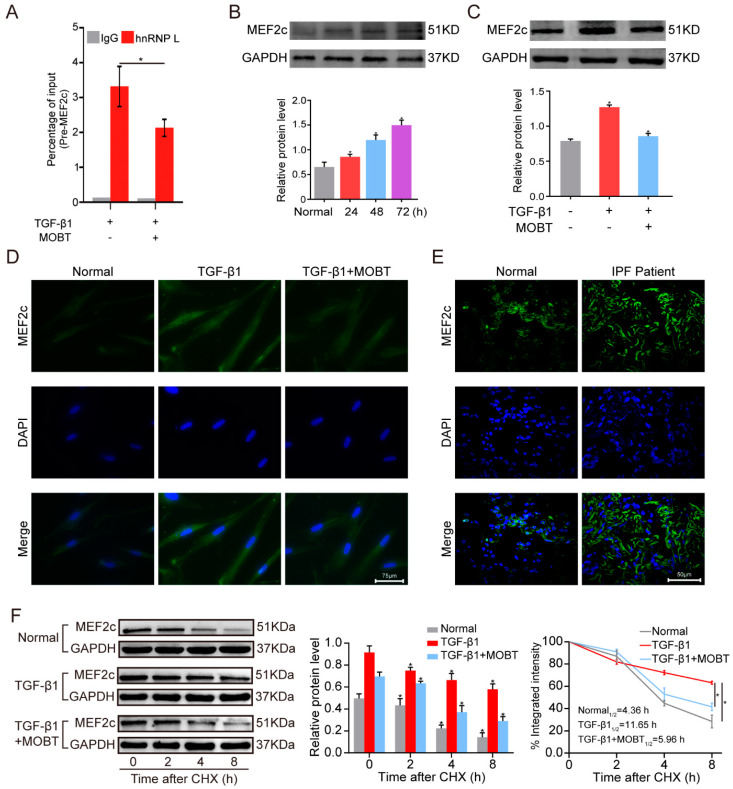
MOBT blocked lung fibrogenesis via inhibiting MEF2c stability. (**A**) The RIP assay showed that MOBT reduced the combination of pre-mMEF2c with hnRNP L. IgG was used as a control. (**B**) The Western blot analysis revealed that MEF2c expression increased in the TGF-β1-treated cells for different time points. (**C**) The Western blot analysis showed that 10 µg/mL MOBT decreased the MEF2c expression. (**D**) The immunofluorescence result demonstrated that the TGF-β1 group expressed abundant MEF2c compared with the normal group, whereas the TGF-β1 + MOBT group decreased the MEF2c expression. The green color indicates MEF2c. The blue color indicates the nucleus. (**E**) The immunofluorescence images showed that the MEF2c expression increased in IPF patient’s lung tissues compared with the normal tissue adjacent to lung cancer. The green color indicates MEF2c. The blue color indicates the nucleus. (**F**) Stability testing showed that MEF2c stability was promoted by TGF-β1 and decreased by MOBT treatment. The half-life in each group was 4.36, 11.65, and 5.96 h in the normal, TGF-β1, and TGF-β1 + MOBT groups, respectively. Each bar represents mean ± SD, * *p* < 0.05.

**Figure 5 molecules-27-05336-f005:**
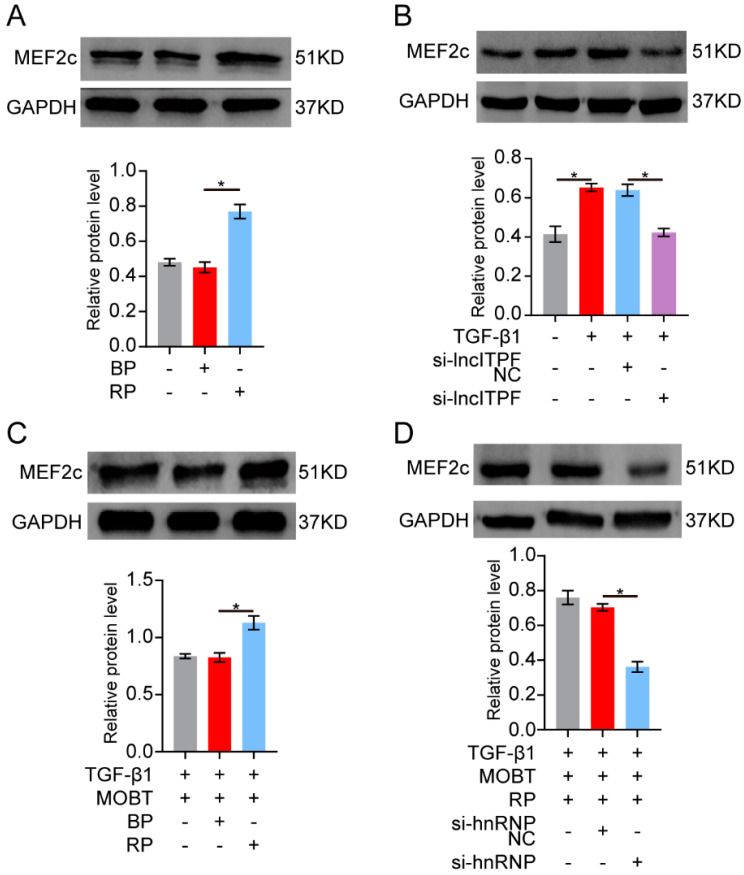
MOBT downregulated MEF2c expression through the lncITPF–hnRNP L complex. (**A**) The Western blot analysis showed that lncITPF overexpression promoted the expression level of MEF2c. (**B**) The Western blot analysis showed that si-lncITPF reduced the expression level of MEF2c. (**C**) The rescue experiments discovered that overexpression of lncITPF reversed the MOBT function, resulting in the blockage of MEF2c expression. (**D**) The rescue experiments demonstrated that si-hnRNP L reversed the joint action of MOBT and overexpression of lncITPF, resulting in a repressed MEF2c level. NC meant the negative control, BP meant the blank plasmid, and RP meant the overexpressed lncITPF recombinant plasmid. Each bar represents mean ± SD, * *p* < 0.05.

**Figure 6 molecules-27-05336-f006:**
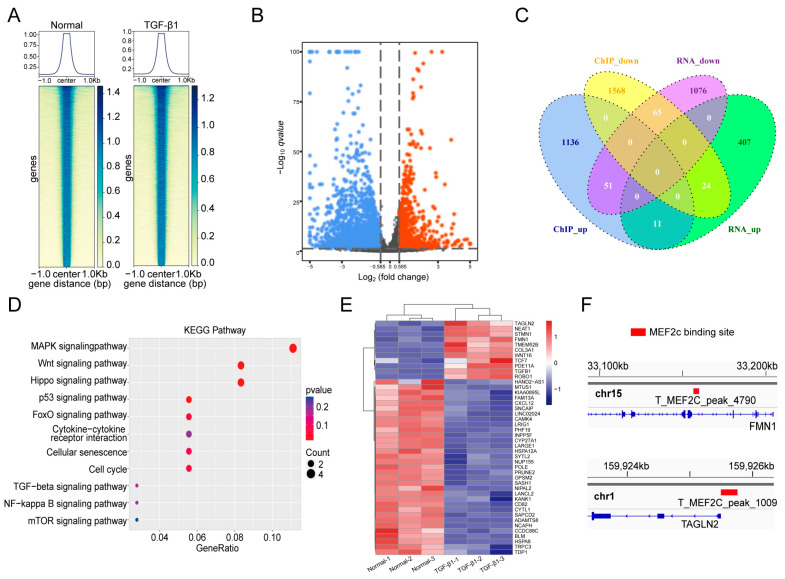
MOBT regulated the downstream target genes of MEF2c. (**A**) A ChIP-seq experiment was performed to identify genes binding with MEF2c. (**B**) RNA-seq was performed to analyze the differentially expressed genes. (**C**) Combined analysis revealed a total of 76 target genes in the ChIP-seq and RNA-seq data, with 11 upregulated and 65 downregulated genes. (**D**) The KEGG analysis revealed that the 76 target genes regulated by MEF2c were significantly enriched in the Hippo, Wnt, and MAPK signaling pathways. (**E**) The 44 significant differentially expressed genes that were selected from 76 target genes and listed in the hierarchical clustering of RNA-seq. (**F**) The binding sites for MEF2c in the TAGLN2 and FMN1 genes were analyzed.

**Figure 7 molecules-27-05336-f007:**
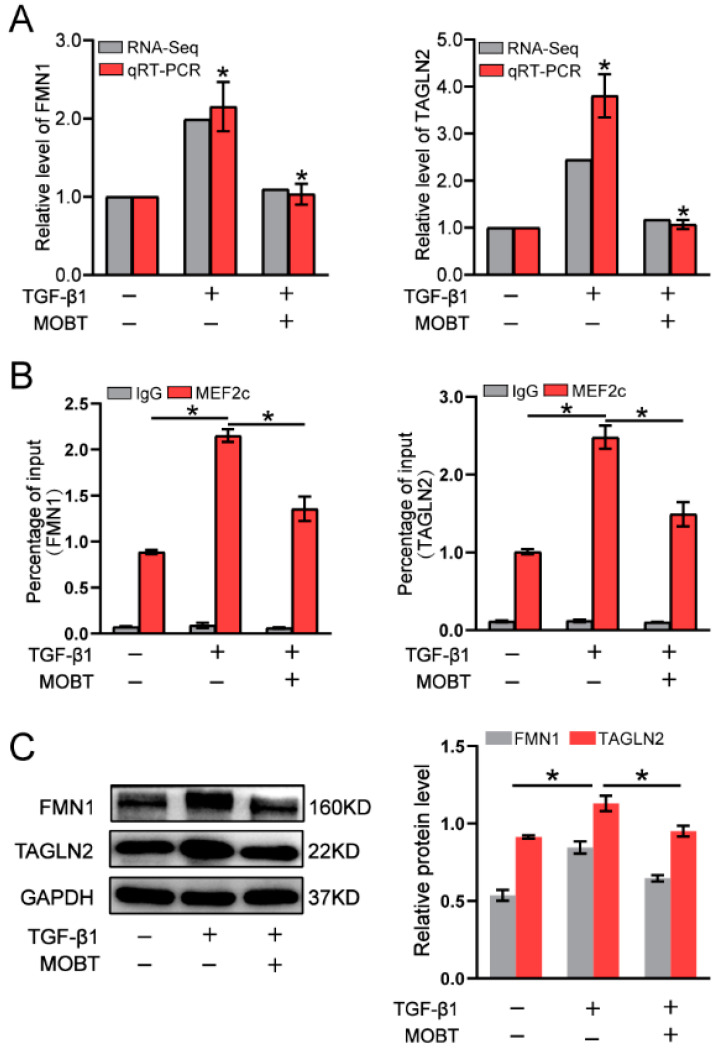
MOBT regulated TAGLN2 and FMN1 via MEF2c. (**A**) The qRT-PCR and RNA-seq data demonstrated that TAGLN2 and FMN1 decreased under MOBT treatment compared with those in the TGF-β1 group. (**B**) The RIP experiment demonstrated that MEF2c bound to TAGLN2 and FMN1. (**C**) The Western blot result verified that MOBT inhibited TAGLN2 and FMN1. Each bar represents mean ± SD, * *p* < 0.05.

**Figure 8 molecules-27-05336-f008:**
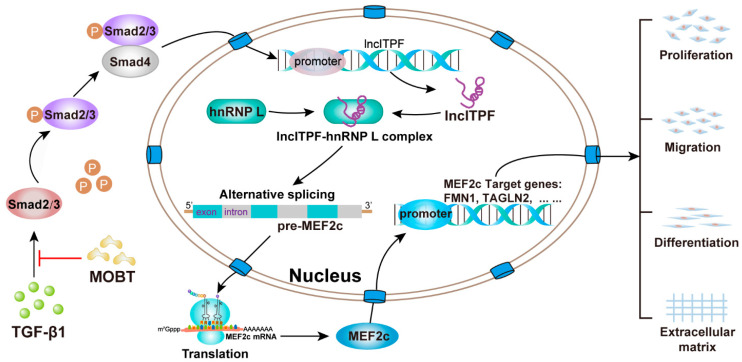
Anti-pulmonary fibrotic mechanism of MOBT. MOBT inhibited lncITPF transcription by preventing p-Smad2/3 translocation from the cytoplasm to the nucleus, resulting in the reduction of the lncITPF–hnRNP L complex. The decreased lncITPF–hnRNP L complex reduced MEF2c expression by blocking its alternative splicing, which in turn inhibited the expression of MEF2c target genes, such as TAGLN2 and FMN1.

**Table 1 molecules-27-05336-t001:** The item number and the brand of antibody.

Antibody Name	Item Number	Brand of the Antibody
GAPDH	AF7021	Affinity
a-SMA	Ab7817	Abcam
Vimentin	AF7013	Affinity
Collagen I	AF7001	Affinity
MEF2c	Ab78888	Abcam
hnRNP L	Ab6106	Abcam
Smad2/3	8685	Cell Signaling Technology
p-Smad2/3	8828	Cell Signaling Technology
ADAMTS8	Bs-5859R	Bioss
TAGLN2	15508-1-AP	Proteintech
HSPA12A	Ab200838	Abcam
FMN1	25982-1-AP	Proteintech

**Table 2 molecules-27-05336-t002:** The primers used in experiments.

Experimental Type	Gene Name	Primer Sequence (5′ to 3′)
qRT-PCR	lncITPF	F: ACCGCCTAATACGACTCACTATAGGGACCC CACCATGGACTCAGTGATAGGAACAAAATGT R: TTTTTTTTTTTTTTTTTTTTTTTTTTTTTTACT AACAT AAAACAGTGTGCTAATCA
qRT-PCR	TAGLN2	F: CCTTCAAGCAGATGGAGCAG R: CTTCCCAGAGGTCCACAGTT
qRT-PCR	FMN1	F: ATGTGGATGACTCCGTGGTT R: GATTATGCACTGGGCACGTT
ChIP-PCR	lncITPF	F: GAAAAAGCCCTCACAAAAGCCTCAC R: GGGGAGGTTACTTTGTGGAAGGATC
ChIP-PCR	TAGLN2	F: CTCTGGAAGCACGCCTTTG R: ATCTATCTGTGAGGGTCCTGAG
ChIP-PCR	FMN1	F: TTAGCTGGACATGGTGGTG R: TCACAATATTAGCCAAACACTG
RIP	pre-MEF2c	F: ACCGACATGGACAAAGTGCTTCTCA R: ATGTGTGTATGTGTGTGTGGCAGGG

## Data Availability

Not applicable.

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
