# Peer review of "MOBT Alleviates Pulmonary Fibrosis through an lncITPF–hnRNP-l-Complex-Mediated Signaling Pathway"

_molecules, 2022, doi:10.3390/molecules27165336_

Round 1
Reviewer 1 Report
Authors need to justify the following issues:
1- There are a lot of abbreviations in the manuscript, so a section for abbreviations should be added after the abstract.
2- Why authors did not compare lung biopsy from IPF patients with that from normal subjects?
3- The effect of MOBT alone in vivo experiment in rats has not been performed, why?
4- It is not clear if saline was sprayed into the lung in sham group as BLM.
5- In the animals model and ethics statement under materials and methods, line 359"MOBT every day13" what does the number 13 refer to?
6- The manufacturing country for kits and equipment should be mentioned whenever possible.
Author Response
Authors need to justify the following issues:
1- There are a lot of abbreviations in the manuscript, so a section for abbreviations should be added after the abstract.
Answer: Thank you for the instructive suggestion. We have added the abbreviations section after the abstract. The revised manuscript as follows:
Abbreviations: BLM: bleomycin, ChIP: chromatin immunoprecipitation, CTGF: connective tissue growth factor, FMN1: formin 1, FVC: forced vital capacity, IPF: idiopathic pulmonary fibrosis, lncRNA: long noncoding RNA, MEF2c: myocyte enhancer factor-2c, MOBT: 3-(4-methoxyphenyl)-4-oxo-4H-1-benzopyran-7-yl(αS)-α,3,4-trihydroxybenzenepropanoate, qRT-PCR: quantitative real time polymerase chain reaction, RIP: RNA-binding protein immunoprecipitation, RNA-FISH: RNA fluorescence in situ hybridization, RTCA: real-time cellular analysis, TAGLN2: transgelin 2, TGF-β1: transforming growth factor-β1, VEGF: vascular endothelial growth factor.
2- Why authors did not compare lung biopsy from IPF patients with that from normal subjects?
Answer: Thank you for the instructive suggestion. We made a comparison of lung biopsy between IPF patients and normal subjects. The revised manuscript as follows:
The MEF2c expression was increased in the IPF patient’s pulmonary tissues compared with the normal subjects (Figure 4E), indicating that the study of MOBT targeted MEF2c has clinical value in medical treatment.
3- The effect of MOBT alone in vivo experiment in rats has not been performed, why?
Answer: Thank you for the instructive suggestion. We have performed the vivo experiment in mice of MOBT alone and found that MOBT alone had no toxicity to the lung tissue of mice. This part of the result had been added in the revised manuscript as follows:
The MicroCT imaging system for small animals demonstrated that both lungs of mice in the BLM group had diffuse reticular blurring, while MOBT alone had no obvious changes compared with the sham group. And the BLM+MOBT treatment group significantly reduced the lesion range of reticular changes compared with the BLM group. (Figure 1B).
Figure 1. (B) The MicroCT imaging system for small animals displayed that the plain scan of the lung window in the sham group showed the texture of both lungs was clear, and no exudation or space occupying lesions were found in the lung parenchyma. The MOBT alone had no obvious changes compared with the sham group. However, both lower lungs of the mice in the BLM group showed diffused reticular blurring and interstitial changes. Meanwhile, the BLM+MOBT treatment group significantly reduced the lesion range of reticular changes.
H&E and Masson staining revealed that the BLM group had more collagen deposition and a thicker alveolar wall than the sham group, while MOBT alone group was similar to the sham. And the BLM+MOBT treatment attenuated collagen deposition and had a significant therapeutic effect (Figure 1E).
Figure 1. (E) H&E and Masson staining demonstrated that a portion of the alveolar structural damaged, alveolar fusion, and collagen fibers gathered in the lung of the BLM mice compared with the sham group. While MOBT alone group was similar to the sham. The BLM+MOBT treatment improved the alveolar structure and attenuated collagen deposition.
4- It is not clear if saline was sprayed into the lung in sham group as BLM.
Answer: Thank you for the instructive suggestion. The sham group was sprayed the same dose of normal saline into the lung as BLM. The revised manuscript in Materials and Methods as follows:
4.2 Animal model and ethics statement
C57BL/6 male mice were purchased from Jinan Pengyue Experimental Animal Breeding Company (Jinan, China). The mice were randomly divided into sham, BLM and BLM+MOBT groups (10 mice in each group). 5 mg/kg BLM were sprayed into the lungs of mice by a Penn-Century MicroSprayer (Penn-Century Inc., USA) in the BLM and BLM+MOBT groups. From the eighth day, the BLM+MOBT group was orally administered 30 mg/kg MOBT every day. The sham group was sprayed the same dose of normal saline into the lung as BLM. On the 28th day, the lung changes of all mice were evaluated by a MicroCT imaging system for small animals (PerkinElmer, USA). Then, lung samples were collected, processed and stored for the following experiments. Experiments on the animals were approved by the Animal Experiments Ethics Committee of Binzhou Medical University (Ethical Research approval No. 2017-02).
5- In the animals model and ethics statement under materials and methods, line 359"MOBT every day13" what does the number 13 refer to?
Answer: The number 13 is misplaced and it should be the original reference and we deleted it in the submitted manuscript. Thank you for pointing out our mistakes so that we can corrected it. We are sorry to make such a mistake. The revised manuscript as follows:
4.2 Animal model and ethics statement
C57BL/6 male mice were purchased from Jinan Pengyue Experimental Animal Breeding Company (Jinan, China). The mice were randomly divided into sham, BLM and BLM+MOBT groups (10 mice in each group). 5 mg/kg BLM were sprayed into the lungs of mice by a Penn-Century MicroSprayer (Penn-Century Inc., USA) in the BLM and BLM+MOBT groups. From the eighth day, the BLM+MOBT group was orally administered 30 mg/kg MOBT every day. The sham group was sprayed the same dose of normal saline into the lung as BLM. On the 28th day, the lung changes of all mice were evaluated by a MicroCT imaging system for small animals (PerkinElmer, USA). Then, lung samples were collected, processed and stored for the following experiments. Experiments on the animals were approved by the Animal Experiments Ethics Committee of Binzhou Medical University (Ethical Research approval No. 2017-02).
6- The manufacturing country for kits and equipment should be mentioned whenever possible.
Answer: Thank you for the instructive suggestion. We added the manufacturing country for kits and equipment in Materials and Methods. The revised Materials and Methods as follows:
- Materials and Methods
4.3 Cell model and treatment
Human embryonic lung fibroblast MRC-5 cell line was purchased from American Type Culture Collection (ATCC, CCL-171™, USA). The cells were cultured in advanced minimum essential medium (Gibco, 11090081, USA) containing 10% FBS (Gibco, 10270106, USA) in a 37°C, 5% CO2 incubator. Cell samples were divided into three groups: normal, TGF-β1 and TGF-β1+MOBT groups. The TGF-β1+MOBT group was first treated with 5 ng/mL TGF-β1 (Gibco, PHG9202, USA) for 24 h and then co-treated with 10 μg/mL MOBT for 48 h
4.5 H&E and Masson Staining
The harvested lung tissues were immediately fixed in 4% paraformaldehyde (Meilunbio, MA0192, China) overnight, then dehydrated, embedded in paraffin. Sections (4 μm) were stained with H&E (Novland, IH-017 and IH-018, China) or a modified Masson’s trichrome staining kit (Solarbio, G1345, China). Images were obtained under the microscope.
4.6 Immunofluorescence observation
Lung tissue paraffin sections were dewaxed according to the manufacturer’s instruction, incubated with 0.3% TritonX-100 (Sinopharm, 30188928, China) for 8 min, blocked with 10% goat serum (Solarbio, SL038, China) for 50 min, then incubated with the antibody of α-SMA (1:200, Abcam, Ab7817, USA) at 4°C overnight. On the second day, sections were rinsed with 1×PBS solution (Sparkjade, CR0013, China) and incubated with anti-IgG Fluor (1:200, Affinity, S0006, China) for 1 h. The nuclear was stained with DAPI (1:400, Sigma, D9542, USA). After 1×PBS washing, tissue sections were sealed with neutral glycerin and took photoes in automatic living cell fluorescence microscopic imaging system (Invitrogen, EVOS M5000, USA).
4.7 Half-life of lncITPF analysis
Cells (1×106/mL) were seeded in a 6 cm petri dish. 2 mL serum-free medium containing 5 μg/mL actinomycin D (Aladdin, A113142, China) was added into the cell samples. Total RNA was extracted and lncITPF levels were measured by qRT-PCR at different time points. GAPDH was used as an internal standard.
4.8 ChIP-PCR
ChIP-PCR assay was performed by using the SimpleChIP® enzymatic chromatin IP kit (Cell Signaling Technology, #9002, USA) according to the manufacturer’s instructions. Briefly, cell samples were crosslinked with 1% formaldehyde (Macklin, F809702, China) for 10 min at room temperature. Then, 2.5 M glycine (Meilunbio, MB4166, China) was added to terminate crosslinking for 5 min. Chromatin was immunoprecipitated with anti-Smad2/3 (Cell signaling technology, 8685, USA) or IgG (Cell signaling technology, 2729, USA) overnight. The antibody/antigen complexes were recovered with protein G agarose beads (Cell signaling technology, 9007, USA) for 2 h at 4°C. After two sequential elutions, 0.2 M NaCl was added to the eluent at 65°C overnight for decross-linked. The immunoprecipitated DNA was collected and then tested by PCR. The primers were listed in Table 2.
4.9 RNA-FISH
lncITPF, U6 and 18S FISH probes were synthesized by Guangzhou RiboBio Co., Ltd. The experiment was performed with the Ribo lncRNA FISH Probe Mix according to the manufacturer’s protocol (RiboBio, C10910, China). Cells were inoculated on a circular slide. When the cell density reached 50–60%, 4% paraformaldehyde (Meilunbio, MA0192, China) was added. Then, cell samples were washed with 1×PBS (Sparkjade, CR0013, China), and punched with 0.3% TritonX-100 (Sinopharm, 30188928, China) at 4°C for 3min. Permeate solution was washed away with 1×PBS and 200 μL pre-hybridization solution was added for 30 min. Subsequently, lncITPF, U6 and 18S FISH Probe Mix were added to the cell samples in a hybridization oven at 37°C overnight. On the second day, the hybridization probe solution was aspirated at 42°C and washed with SSC (Solarbio, S1030, China). Then DAPI (Sigma, D9542, USA) solution was added for 6 minutes. Finally, fluorescence was observed by a laser confocal microscope.
4.10 RTCA
Cells (5×104/mL) were seeded in E-plate (Agilent, 5469830001, USA) for proliferation analysis and CIM plate (Agilent, 05665817001, USA) for migration analysis, respectively. The CIM plate contains the upper chamber and the lower chamber. 30 μL serum-free medium and 165 μL serum medium was added into the upper or lower chamber, respectively. The RTCA instrument automatically recorded the proliferation or migration curves. The cell index was calculated by RTCA software (ACEA Biosciences, USA).
4.11 RIP analysis
RIP assay was performed using an EZ-Magna RIP™ RNA-binding protein immunoprecipitation kit (Millipore, 17-701, USA) according to the manufacturer’s instructions. Firstly, RIP lysis buffer was added to cell samples. Then hnRNP-L antibody (Abcam, ab264340, USA) or IgG antibody (Cell Signaling Technology, 2729, USA) conjugating with magnetic beads was put in the whole cell extract at 4°C overnight. Next, beads were collected after incubation with protein A/G at 4°C for 2 h and washed 5 times. Purified RNA was analyzed by qRT-PCR with MEF2c pre-mRNA specific primers. The primers were listed in Table 2.
4.13 ChIP-Seq
ChIP assay was performed by Shandong Xiuyue Biotechnology Co., Ltd, according to the standard crosslinking ChIP protocol with modifications. Briefly, cells were harvested and crosslinker with 1% formaldehyde for 10 min at room temperature. After sonication, immunoprecipitation was performed with anti-MEF2c (Santa, sc-518152, 10 μL, USA). The immunoprecipitated complex was washed, and DNA was extracted and purified by Universal DNA Purification Kit (TIANGEN, #DP214, China). The ChIP-Seq library was prepared using original Ultra II DNA Library Kits (NEB, #E7645, USA) according to the manufacturer’s instructions. For ChIP-seq, extracted DNA was ligated to specific adaptors followed by deep sequencing in the Illumina Novaseq 6000 using 150bp paired-end.

Reviewer 2 Report
I have two questions:
1. I will be interested in the results of whether the author is using EPR or other analyses to present their thesis.
2. Does they have results about the other cytokines?
Recomendation
The article is innovative and really interesting for medicine. The section" discussion" needs to be expanded!
Author Response
I have two questions:
- I will be interested in the results of whether the author is using EPR or other analyses to present their thesis.
Answer: Thank you for the instructive suggestion. We did not perform EPR assay to analyse the compound, and we used 1H NMR and LC/MS to identified it, as listed in following. Indeed, EPR experiment is of great significance for the characterization of compounds and the interpretation of their biological effects. We appreciate the reviewer’s excellent question and would like to consider related studies in the future work to explore more solid evidence.
Figure R1. The 1H NMR result of the MOBT compound.
Figure R2. The LC/MS result of the MOBT compound.
- Does they have results about the other cytokines?
Answer: Thank you for the instructive suggestion. We have detected the expression of cytokines related to fibrosis, such as transforming growth factor-β1 (TGF-β1), connective tissue growth factor (CTGF), and vascular endothelial growth factor (VEGF). Western blot results manifested that MOBT distinctly decreased the expression of TGF-β1, CTGF, and VEGF compared with the BLM group. The revised manuscript and figure 1 as follows:
Western blot results manifested that MOBT distinctly decreased the expression of fibrotic markers, including α-SMA, vimentin, and collagen I, compared with the BLM group (Figure 1F). Meanwhile, MOBT also inhibited the expression of cytokines related to fibrosis, such as TGF-β1, CTGF, and VEGF (Figure 1G).
Figure 1. (G) Western blot demonstrated that MOBT inhibited the expression of TGF-β1, CTGF and VEGF, which were cytokines related to fibrosis, compared with those in the BLM group.
Recomendation
The article is innovative and really interesting for medicine. The section" discussion" needs to be expanded!
Answer: Thank you for the instructive suggestion. We expanded the section of discussion. The revised manuscript as follows:
In recent years, lncRNAs receive increasing attention in the fields of life science, medicine, and pharmacy because of their conservation and high tissue-specificity [22-24]. Cis-acting lncRNAs regulate genes expression which were located on the same chromosome, while trans-acting lncRNAs regulate genes on the other chromosomes [25]. They play important functions in the spatial conformation of chromosomes, RNA transcription, pre-mRNA splicing, mRNA degradation, and translation by interacting with proteins [13,26]. For example, the lncIAPF–HuR complex inhibits autophagosome fusion with lysosome to block autophagy, resulting in pulmonary fibrogenesis [27]. lncHOTAIR regulates tumor cell proliferation by binding to Y-Box Protein-1 and promoting its nuclear translocation, which stimulating Y-Box Protein-1 downstream targets PCK2 and PDGFRβ [28]. Accordingly, lncRNAs have become therapeutic targets in many diseases, such as pulmonary, renal, liver, and cardiac fibrosis [29,30]. lncMER52A can serve as a diagnostic and prognostic marker for hepatocellular carcinoma [31]. lncRAPIA may become a new prevention and treatment target for advanced atherosclerosis [32]. However, these studies mostly focused on gene therapy and paid little attention to drug treatment. Numerous herbal compounds, such as curcumin, camptothcin, resveratrol, quercetin, genistein, 3,3′-diindolylmethane, and epigallocatechin-3-galate, exhibit anti-cancer effects through controlling the lncRNA expression [33]. The lncRNA-mRNA ChIP analysis uncovered the action of Caulophyllum robustum Maxim, which can play a part in rheumatoid arthritis via lncRNA-mediated signing pathways, including tumor necrosis factor, chemokine, and Toll-like receptor signaling pathways [34]. These studies simply screened the differentially expressed lncRNAs under the action of drugs. However, the drug mechanism mediated by lncRNA remains obscured. This work found that MOBT blocked MEF2c alternative splicing to prevent pulmonary fibrosis by reducing the formation of lncITPF–hnRNP L complex, and the lncITPF–hnRNP L can be the therapeutic drug target in pulmonary fibrosis.
... ...
MEF2c belongs to the MEF2 transcription factor family, which is expressed in a variety of cells and associated with neurodevelopmental disease, heart failure, and vascular inflammation [38-40]. MEF2c initiates the transcription of target genes and induces the cells to develop a series of adaptive response. It can direct fibroblast reprogramming and correlate with fibrosis in vivo. According to Zhang et al., salvianolic acid B prevents hepatic stellate cell activation and hepatic fibrogenesis by down-regulating the MEF2c signaling pathway [41]. Our RIP, qRT-PCR, and western blot data confirmed that MEF2c bound to the promoter of TAGLN2 and FMN1 and promoted the transcription of these target genes. These target genes belong to the Hippo, Wnt, and MAPK signaling pathway, which can activate myofibroblast differentiation, proliferation, migration, and ECM deposition to promote pulmonary fibrosis. Meanwhile, MOBT treatment inhibited TAGLN2 and FMN1 through downregulating the lncITPF–hnRNP L complex-targeted MEF2c.
References
[25] Shaath, H.; Vishnubalaji, R.; Elango, R.; Kardousha, A.; Islam, Z.; Qureshi, R.; Alam, T.; Kolatkar, P. R.; Alajez, N. M. Long non-coding RNA and RNA-binding protein interactions in cancer: Experimental and machine learning approaches. Semin Cancer Biol. 2022, S1044-579X(22)00124–9.

Reviewer 3 Report
In the present article, the authors review the therapeutic potential of the anti-fibrotic therapeutic3-(4-methoxyphenyl)-4-oxo-4H-1-benzopyran-7-yl(αS)-α,3,4 trihydroxybenzenepropanoate (MOBT) in a mice and cell model of bleomycin-induced and transforming growth factor-β1 lung parenchymal injury. In the study, convincing results are presented, showing that MOBT alleviates lung fibrosis through the lncITPF–hnRNP L complex targeting the MEF2c signaling pathway. The authors reveal the possible application of MOBT in the development of new strategies for the treatment of pulmonary fibrosis.
It put well the paper’s objective, clear and sufficiently short. The manuscript is at a high scientific level and structured accordingly to the instructions of the journal. The design of the study and methodology is appropriate to the paper topic.
Results are described in detail and discussed in the correct volume. The section on results does deal exclusively with results. The experimental details are very well defined. The discussion and conclusion seem interesting and are scientifically supported in this article. The text is written in a scientific style appropriate for the journal, but it needs to moderate the English changes required. Please correct the entire text according to grammar and spelling rules.
The visibility and quality of figures and tables: are informative and clearly emphasize the significance of the study.
Author Response
In the present article, the authors review the therapeutic potential of the anti-fibrotic therapeutic3-(4-methoxyphenyl)-4-oxo-4H-1-benzopyran-7-yl(αS)-α,3,4 trihydroxybenzenepropanoate (MOBT) in a mice and cell model of bleomycin-induced and transforming growth factor-β1 lung parenchymal injury. In the study, convincing results are presented, showing that MOBT alleviates lung fibrosis through the lncITPF–hnRNP L complex targeting the MEF2c signaling pathway. The authors reveal the possible application of MOBT in the development of new strategies for the treatment of pulmonary fibrosis.
It put well the paper’s objective, clear and sufficiently short. The manuscript is at a high scientific level and structured accordingly to the instructions of the journal. The design of the study and methodology is appropriate to the paper topic.
Results are described in detail and discussed in the correct volume. The section on results does deal exclusively with results. The experimental details are very well defined. The discussion and conclusion seem interesting and are scientifically supported in this article. The text is written in a scientific style appropriate for the journal, but it needs to moderate the English changes required. Please correct the entire text according to grammar and spelling rules.
The visibility and quality of figures and tables: are informative and clearly emphasize the significance of the study.
Answer: Thank you for the instructive suggestion. We have corrected the entire manuscript according to grammar and spelling rules. The revised manuscript as follows. For example:
- Results
2.1. MOBT prevented BLM-induced lung fibrosis in mice
Animal experiments were performed to evaluate the anti-fibrotic effect MOBT on BLM-induced pulmonary fibrosis in mice (Figure 1A). The MicroCT imaging system for small animals demonstrated that both lungs of mice in the BLM group had diffuse reticular blurring, while MOBT alone had no obvious changes compared with the sham group. And the BLM+MOBT treatment group significantly reduced the lesion range of reticular changes compared with the BLM group. (Figure 1B). Weight monitoring showed that MOBT effectively slowed down the weight loss of BLM-treated mice (Figure 1C). Lung function analysis demonstrated that the MOBT treatment improved the FVC compared with the BLM-treated mice (Figure 1D). H&E and Masson staining revealed that the BLM group had more collagen deposition and a thicker alveolar wall than the sham group, while MOBT alone group was similar to the sham. And the BLM+MOBT treatment attenuated collagen deposition and had a significant therapeutic effect (Figure 1E). Western blot result manifested that MOBT distinctly decreased the expression of fibrotic markers, including α-SMA, vimentin, and collagen I, compared with the BLM group (Figure 1F). Meanwhile, MOBT also inhibited the expression of cytokines related to fibrosis, such as TGF-β1, CTGF, and VEGF (Figure 1G). Immunofluorescence and immunohistochemistry results confirmed that the lung mesenchyme was positive staining for α-SMA in the BLM group, indicating the appearance of fibroblastic foci. The MOBT treatment reduced the α-SMA expression and improved the fibrotic status (Figure 2). The above-mentioned results indicated the anti-pulmonary fibrotic ability of MOBT in vivo.
- Discussion
In recent years, lncRNAs receive increasing attention in the fields of life science, medicine, and pharmacy because of their conservation and high tissue-specificity [22-24]. Cis-acting lncRNAs regulate genes expression which were located on the same chromosome, while trans-acting lncRNAs regulate genes on the other chromosomes [25]. They play important functions in the spatial conformation of chromosomes, RNA transcription, pre-mRNA splicing, mRNA degradation, and translation by interacting with proteins [13,26]. For example, the lncIAPF–HuR complex inhibits autophagosome fusion with lysosome to block autophagy, resulting in pulmonary fibrogenesis [27]. lncHOTAIR regulates tumor cell proliferation by binding to Y-Box Protein-1 and promoting its nuclear translocation, which stimulating Y-Box Protein-1 downstream targets PCK2 and PDGFRβ [28]. Accordingly, lncRNAs have become therapeutic targets in many diseases, such as pulmonary, renal, liver, and cardiac fibrosis [29,30]. lncMER52A can serve as a diagnostic and prognostic marker for hepatocellular carcinoma [31]. lncRAPIA may become a new prevention and treatment target for advanced atherosclerosis [32]. However, these studies mostly focused on gene therapy and paid little attention to drug treatment. Numerous herbal compounds, such as curcumin, camptothcin, resveratrol, quercetin, genistein, 3,3′-diindolylmethane, and epigallocatechin-3-galate, exhibit anti-cancer effects through controlling the lncRNA expression [33]. The lncRNA-mRNA ChIP analysis uncovered the action of Caulophyllum robustum Maxim, which can play a part in rheumatoid arthritis via lncRNA-mediated signing pathways, including tumor necrosis factor, chemokine, and Toll-like receptor signaling pathways [34]. These studies simply screened the differentially expressed lncRNAs under the action of drugs. However, the drug mechanism mediated by lncRNA remains obscured. This work found that MOBT blocked MEF2c alternative splicing to prevent pulmonary fibrosis by reducing the formation of lncITPF–hnRNP L complex, and the lncITPF–hnRNP L can be the therapeutic drug target in pulmonary fibrosis.
... ...
MEF2c belongs to the MEF2 transcription factor family, which is expressed in a variety of cells and associated with neurodevelopmental disease, heart failure, and vascular inflammation [38-40]. MEF2c initiates the transcription of target genes and induces the cells to develop a series of adaptive response. It can direct fibroblast reprogramming and correlate with fibrosis in vivo. According to Zhang et al., salvianolic acid B prevents hepatic stellate cell activation and hepatic fibrogenesis by down-regulating the MEF2c signaling pathway [41]. Our RIP, qRT-PCR, and western blot data confirmed that MEF2c bound to the promoter of TAGLN2 and FMN1 and promoted the transcription of these target genes. These target genes belong to the Hippo, Wnt, and MAPK signaling pathway, which can activate myofibroblast differentiation, proliferation, migration, and ECM deposition to promote pulmonary fibrosis. Meanwhile, MOBT treatment inhibited TAGLN2 and FMN1 through downregulating the lncITPF–hnRNP L complex-targeted MEF2c.
Figure legends
Figure 1 (B) The MicroCT imaging system for small animals displayed that the plain scan of the lung window in the sham group showed the texture of both lungs was clear, and no exudation or space occupying lesions were found in the lung parenchyma. The MOBT alone had no obvious changes compared with the sham group. However, both lower lungs of the mice in the BLM group showed diffused reticular blurring and interstitial changes. Meanwhile, the BLM+MOBT treatment group significantly reduced the lesion range of reticular changes. (E) H&E and Masson staining demonstrated that a portion of the alveolar structural damaged, alveolar fusion, and collagen fibers gathered in the lung of the BLM mice compared with the sham group. While MOBT alone group was similar to the sham. The BLM+MOBT treatment improved the alveolar structure and attenuated collagen deposition. (G) Western blot demonstrated that MOBT inhibited the expression of TGF-β1, CTGF and VEGF, which were cytokines related to fibrosis, compared with those in the BLM group.